# Female Empowerment and Radical Empathy for the Sustainability of Creative Industries: The Case of K-Pop

Ingyu Oh [1,*] , Kyeong-Jun Kim [2] and Chris Rowley [3,4]

1   Faculty of Global Engagement, Kansai Gaidai University, Osaka 573-1001, Japan
2   Department of Urban Sociology, University of Seoul, Seoul 02592, Republic of Korea
3   Kellogg College, Oxford University, Oxford OX2 6PN, UK
4   Bayes School of Business, City, University of London, London EC1V 0HB, UK
*   Correspondence: oingyu@kansaigaidai.ac.jp

**Abstract:** Contrary to the critical understanding of the cultural and/or creative industry that unduly emphasizes demoralized commercial activities of profit-maximizing, accompanied by the concomitant destruction of individual self-realization, the 21st century perception of the industry highlights its potentials for both creativity and more importantly, sustainability. The global success of Korean pop music (K-pop) unlocks a new possibility for the creative industry in a postcolonial country, with a newly constructed value chain that strategically allows female empowerment through radical empathy, a concept that bridges social empathy among formerly oppressed groups with their new political opportunities of political struggles. Based on survey data and structural equation modeling, this paper empirically corroborates a theorized conceptual link between participation in the K-pop industry and the resulting radical empathy among female fans toward industrial sustainability.

**Keywords:** sustainability; female empowerment; radical empathy; K-pop; cultural industry; creative industry





## 1. Introduction

In the context of the overarching importance of sustainability across a number of disciplines, the aim of this paper is to rescue the definition of the creative industry from a possible demise, along with its conventional term, "cultural industry," which has often been criticized as a mere profit-making commercial activity that would eventually eradicate human creativity that was once considered a pivotal part of the grand narrative of "self-realization," widely propagated by Marxists and cultural theorists [1–5]. To achieve this purpose, we ask three broad questions: (1) if the creative industry can be anchored in radical empathy for industrial sustainability, a new concept in social psychology that highlights the politically motivated empathetic participation by cultural creators and consumers for class, race, and gender struggles; (2) if the Korean pop music, or K-pop, can realize its desire of sustainability as a creative industry through radical empathy rampant among its creators, artists, and fans; and (3) if we can conceptualize and operationalize the whole industrial process of sustainability into testable hypotheses.

Discovering the potential of the creative industry that would enable the cultural participants in the industry to realize their creative desire, along with the hope of sustaining the industrial growth and profitability, is quintessential. This is particularly true in the business of creating and managing a middlebrow cultural genre that would enhance human creativity, with avant-garde penchants toward artistic rebellions against the mainstream and establishment (or cultural innovation in a radical form), while concomitantly realizing economic sustainability by generating surplus [6,7]. Thus far, only few pop music genres have succeeded in creating and sustaining middlebrow genres, such as jazz music, classic rock, hip-hop, and K-pop [7–11].

The common denominators in these successful middlebrow pop music genres are: (1) race, (2) gender, and (3) postcoloniality. All these music genres came from African music, as jazz was an iconic cultural symbol found among African Americans, just as much as hip-hop came from Jamaica and blossomed commercially in the metropolitan areas of North America [7,10]. Likewise, although classic rock and K-pop are white and Asian pop music genres, respectively, they both have relentlessly borrowed musical elements from jazz, blues, and hip-hop, and hybridized them with their musical traditions [9,12]. While jazz, classic rock, and hip-hop represent masculinity predominantly, K-pop on the other hand symbolizes femininity and/or androgynous masculinity [7,11]. Finally, hip-hop and K-pop are from postcolonial societies, whereas jazz contains colonial messages about slaves and/or racially oppressed "peoples," from Africa.

As we pointed out earlier, these music genres have enjoyed a middlebrow status for their musical innovations and commercial success. What has sustained their legacy as successful artistic and commercial achievements is another commonality of radical empathy rampant among producers, artists, and consumers in the industry. Jazz was tantamount to the formation of the civil rights movements during the 1960s in the U.S., whereas rock music has gained its political momentum through the era of the anti-Vietnam War and peace movement [7,9]. While hip-hop thrived for its role in the fight against colonialism and racism, K-pop gained fame and prosperity for its contribution to the formation of "female universalism" and the politics of gender equality and liberation [10,11]. In these musical communities, radical empathy is a driving force of social change, while race, gender, and postcoloniality constitute the base that serves as a pedestal for the formation of radical empathy [13].

In this paper we, therefore, empirically analyze how race, gender, and postcoloniality affect artistic creativity in the form of self-realization and the formation of radical empathy in a new middlebrow music genre, called K-pop. We argue that gender, race, and postcoloniality influence artistic creativity and facilitate the development of radical empathy not only among artists, but among fans as well. Gender, race, and postcoloniality work as motivating forces for both creation and proactive consumption of K-pop over an extended period, bolstering the industrial sustainability of the creative industry. To substantiate this theoretical complexity, we use K-pop as an empirical case study, with survey data collected from 15 countries. We first review the extant literature to derive hypotheses, followed by a systematic analysis of the survey data, using structural equation modeling. We then present our findings, along with their implications.

## 2. Literature Review and Hypotheses

In this section, we first discuss literature on the creative industry and why it faces an unwarranted theoretical and conceptual demise in cultural studies. We then discuss ways of saving the concept as a congenial industrial option for middlebrow art genres, including K-pop, by focusing on the issue of race, gender, and postcoloniality. We finally analyze the concept of radical empathy as an outcome of the creative industry, which also ensures industrial sustainability. The section will end with derived hypotheses from each theoretical discussion.

### 2.1. Creative Industry

It is well known that it was the Tory government that had coined the term "creative industries", after its victory in the 1997 election [14]. The U.K. government has since maintained that the shift in terminology was motivated by changing theoretical and political interests of the new millennia, which resulted in the legitimacy of the cultural industry [15]. Unlike the precedent concept of the "cultural industries," which has been mostly reduced to the status of profit-making commercialism, this new concept, based on the novel theoretical and political interests, ascertains that legitimacy comes from the new reality that profit-making can go hand in hand with congenial creativity. Indeed, despite heated debates over how to expand or limit the range of its creative activities and productions within the

industry, what was quintessential to this new industry turned out to be twofold: (1) its potential for sustainability via profit-making and innovations; and (2) its impact on social and cultural life through the diversity of talents and creativity [16,17].

The critique of cultural industries stems originally from the Frankfurt School thinkers, who defined the cultural industries as a new devastating force of modernization that would destroy human culture from the demonic outcomes of European and American civilizations, and their cultural industries [3]. The key of the cultural industries is the technological reproducibility of the original, and the mass production and distribution of commercialized art works to the masses. The industry's goal is profit maximization by reducing the masses into single-dimensional crowds, who have simplistic taste that cannot distinguish good from bad art [18], and become all but subservient to the fascist ideology of the Third Reich, or the capitalist propagandas of American-style late capitalism, including infamous advertisements [3].

In recent years, similar critiques are applied to the creative industries for their tendency to repackage their business (or capitalism in general) aesthetically and/or emotionally affective, in order to neuter the cold and heartless image of capitalism that symbolized discriminations against people based on their race, ethnicity, gender, sexuality, class, and other lifestyle patterns. [5,19–22]. Some of these studies nonetheless highlight the difference between aesthetic (emotional) capitalism and traditional capitalism, by arguing that the consumers in late capitalism have become more creative in terms of preferring more aesthetic products over Fordist or post-Fordist commodities (e.g., love images of coffee vs. conventional dinner coffee). Therefore, consumers in late capitalism come to juggle around different types of volatile emotional experiences before and after consumption of such aestheticized products [21,22]. If consumers indeed value emotional satisfactions more than economic utilities, status symbols, and exchange or storage values, the desire of consumption will be not only endless, but it will demand more beautiful and emotionally complex varieties via innovations [22].

However, as Eagleton (2009) explains correctly, these aestheticized and emotional consumers are seemingly treated equally regardless of their economic, racial, gender, and postcolonial specificities, only because capitalists see them having money and being willing to pay for the goods and services. It is not that the system has become fair over time for humanity, but capitalism has evolved into a universal free market mechanism that sees none other than money as a social discriminator [5]. Whereas the chasm between economic classes has only widened due to this late capitalism, social discriminations based on race, gender, and postcoloniality, have seemingly waned from the market where the creative industries operate.

What these scholars are critical about within creative industries is, therefore, their devastating effect on culture itself, as they tend to destroy art and culture, along with individual creativity, a means of individual self-realization, by subjugating diversity and hybridity to the principles of late capitalism that do not discriminate against individuals, as long as they have the money to buy diverse, aesthetic, and emotional products. Without emancipating individuals from the yoke of passive consumerism, the ideal of self-realization, therefore, cannot be achieved.

### 2.2. Race, Gender, and Postcolonial Melancholia

The cultural industries view of the creative industry presents a static and overly deterministic view of the cultural world of late capitalism and globalization, in that most of them present that capitalism, whether conventional, aesthetic, or creative, is detrimental to individualism, self-realization, and class equity. However, they fail to explain why a postcolonial popular culture is now dominating the global music market, even though it is not about capitalist profit-maximizing on a global scale. Even as K-pop is dominating the Japanese pop market as the most popular music genre, it does not mean that Korean capitalism is destroying and controlling its Japanese counterpart. It simply means that Japanese popular culture is now more inclusive and diverse than before, by consuming its

postcolonial pop music. In this sense, our view that emphasizes the importance of race, gender, and postcoloniality in the creative industries discloses the potential of enhancing one's chances of participating to creative activities, even if it entails mass consumptions through the aestheticization of postcolonial pop. What is required in this new type of cultural consumption is the discovery of the new sources of creativity (i.e., a new innovative type of aestheticizing and emotionalizing postcolonial experiences, that can satisfy Japanese or any former colonialist consumers), which will lead to the creation and sustainability of the middlebrow cultural genres (i.e., consumers keep demanding postcolonial pop because of the discovery and addiction to this new type of aesthetic innovation). The core of these sources of creativity is the new perspective in the creative industries that considers race, gender, and postcolonial melancholia seriously. This also means that the success of postcolonial pop culture is not the same as the destruction of individual creativity among the postcolonial artists, nor does it entail the destruction of proactive consumption among conscious consumers in the center countries.

As mentioned earlier, the 21st century popular culture is different from that of the 20th century in that non-white, non-European, or non-male-centered genres are as powerful as their conventional white, European, and male-centered genres, as exemplified by jazz, hip-hop, Bollywood, and K-pop [23–26]. The key to the success of alternative and rebellious genres that constitutes the middlebrow art between European high culture and Hollywood-style pop culture, is the discovery of racial, gendered, and postcolonial melancholia among cultural creators and consumers [11]. This implies that this new type of postcolonial middlebrow art is much more conscious of racial, gender, and postcolonial justice, which emotional capitalism tends to reduce to individual emotional experiences, that would not easily escalate into a collective action. However, postcolonial middlebrow art is fundamentally collective in nature, as the emotion it tries to convey to consumers are consciously anchored in the social conflict toward racial, gender, and postcolonial justice. Having explained the importance of postcolonial art based on race, gender, and postcoloniality in the creative industries of the 21st century, our next question is: what is the motivational factor of creativity among postcolonial artists? Even as most studies that highlight the importance of race and gender in the creative industries rightly acknowledge them as sources of creativity, they fail to pin down the motivational factors of creativity within the realm of race and gender. It is the psychoanalysts, cultural study scholars, feminist philosophers, and neuroscientists who discovered this new connectedness between melancholia and cultural creativity [11]. Three types of melancholia exist, when the term is psychoanalytically defined as the sense of loss and unfulfilled mourning: (1) gendered, (2) racial, and (3) postcolonial [27–31]. Gendered melancholia emanates from the loss of the desired gender amid outright sexism, which cannot be resolved by mourning, while racial melancholia is the outcome of the loss of the desired race amid persistent racism, which also cannot be resolved by mourning [27,28,31]. Postcolonial melancholia, which is the newest ailment in human history, comes from the loss of the desired colony, which leads to informal and unobtrusive hatred of the people from the former colony, who happen to reside in the former colonizer country. Victims of hatred and racism within the land of former colonizers then experience racial melancholia [32]. Among the three, gendered melancholia is the archetypal form, as it is the oldest in its existence, while racial and postcolonial types are outcomes of modernity [27,28].

The correlation between these three types of melancholic depression and artistic creativity, a critical fulcrum in the creation of middlebrow art genres, has been well documented by psychoanalysts [33–36]. Jamieson (1993), for example, emphasized an interesting relationship between manic-depressive disorder and creativity, first pointed out by the German psychiatrist Emil Kraepelin in 1921. Kraepelin was the psychiatrist who first distinguished manic depression from schizophrenia and argued that manic-depressive illness brings about changes in the thought processes, that "set free powers which otherwise are constrained by all kinds of inhibition. Artistic activity . . . may . . . experience a certain furtherance" [34] (p. 55).

Jamieson (1993) also found that artists have traditionally shown a lucidly higher rate of bipolar and unipolar illnesses than non-artists, including celebrated painters such as Vincent van Gogh and Edvard Munch [35] (p. 404). Her contribution to the study of artistic creativity is the finding that living writers are four times more likely to have manic depression and three times more likely to have depression as people who are not creative. This finding is reaffirmed by Akiskal and Akiska (1988), who found that nearly two-thirds of European writers, painters, and sculptors have manic depression and more than half of them had suffered a major depressive incident [35,37]. The underlying shared premise among these psychiatrists, who had spent time studying the creativity of artists, is that people with manic-depressive illness "experience an exhilarating feeling of energy and a capability for formulating ideas that dramatically enhance artistic creativity" [35] (p. 404). According to Jamison (1993), the interaction of tension and transition between changing mood states are critically important, as well as the sustenance and discipline that manic-depressive patients draw from periods of health. These tensions and transitions ultimately give creative power to the artist. These hypotheses surrounding the linkage between depression and creativity have further been corroborated by other researchers with empirical evidence [35,36]. This line of research supports the clinical conviction that a genetic vulnerability to manic-depressive illness might be accompanied by a predisposition to creativity. Richards' interpretation of her findings is that genes associated with a greater risk of manic-depressive disorder may also confer a greater likelihood of creativity. This is not to imply that the illness creates the predisposition to creativity, but rather that people who have this illness also have capabilities such as extreme exuberance—enthusiasm and energy—that express themselves in creativity [35].

Although the study of middlebrow art genres, such as jazz, has not yet incorporated these psychiatric and neuroscientific studies of creativity [7], some previous studies have documented the connection between melancholia and K-pop for both artist (e.g., creation of art works) and fan creativity (e.g., formation of proactive fan activities, creation of cover dance videos, learning behaviors of Korean language, history, culture, etc.) toward artistic values that can be consumed en masse by fans who share at least one or all the three types of melancholia [11]. K-pop's artistic value is to create an aesthetic aspect of gendered, racial, and postcolonial melancholia, while it has also motivated massive and widespread fandom participation toward radical empathy [26,38,39]. Unlike jazz, however, K-pop is deeply rooted in gendered melancholia, as its fans are mostly women, although its artistic messages also encompass racial and postcolonial spectrums [11]. K-pop, therefore, is an exemplary case of the middlebrow art genre that shows a clear linkage between female fans and their creativity in terms of presumption (i.e., proactive consumption in tandem with their own creative activities of producing cover dances, parodying, learning Korean culture and language, and other fandom activities) and radical empathy toward female universalism and gender emancipation.

### 2.3. Radical Empathy

Radical empathy is a new concept that is derived from the conventional idea of empathy, widely discussed both in sociology and social work studies. According to Givens (2021, p. 34), who is a pioneer in empirically establishing radical empathy, the concept has the following elements that are shared partially with the conventional concept of empathy [13]:

- Becoming grounded in who you are
- Opening yourself to the experiences of others
- Practicing empathy
- Taking action
- A willingness to be vulnerable
- Creating change and building trust

The first four elements of radical empathy are common to the conventional conception of empathy, which is an opposite concept to indifference. What stands out as new elements

for being radical are the final two, i.e., a willingness to be vulnerable, and creating change and building trust. Givens (2021, p. 72) further elaborates on these two notions in the following fashion [13]:

- Paying attention to what is going on in your community and the way that you interact with others.
- Being vulnerable means being honest and truthful: are there ways that you have allowed yourself to be vulnerable with yourself and with others around the issue of race (as well as gender and postcoloniality)?
- Do you find that your friends and family feel comfortable expressing their feelings and opinions around you?
- What kind of neighborhood do you live in? Does your neighborhood reflect the diversity of the region where you live? If not, why?
- Do you know people from different backgrounds and cultures? Take the time to listen to people from other cultures and learn more about their life experiences.
- What type of cultural events do you attend? Are you willing to explore events from cultures different from your own?

Being vulnerable, therefore, presupposes an existential confidence about oneself vis-à-vis the danger surrounding his/her true identity, that is often prescribed by gender, race, and postcoloniality. For example, while being a black woman in the U.S. and elsewhere, where racism, sexism, and discriminations against migrants from former colonies are severe, when she tries to preserve and reveal her true identity without succumbing to the force of cultural assimilation, she will face various dangers. Givens (2021, p. 84) cites the following statistics as examples [13]:

- Structural racism, the systems-level factors related to, yet distinct from, interpersonal racism, leads to increased rates of premature death and reduced levels of overall health and well-being.
- Black women are three to four times more likely to experience a pregnancy-related death than White women.
- Black women are more likely to experience preventable maternal death compared with White women.
- Black women's heightened risk of pregnancy-related death spans income and education levels.

The final stage of radical empathy is, therefore, creating change and building trust by being vulnerable, even as the act of being vulnerable includes attending cultural events and exploring events from cultures different to your own. Being radically empathetic is, therefore, almost quintessential to listening to jazz and consuming K-pop as far as they contribute to creating changes and building trust based on a community-wide struggle against racism (jazz, K-pop), sexism (K-pop), and discriminations against migrants (K-pop).

Female fans in the K-pop community are known for their radical empathy of intercultural understanding toward female universalism (i.e., sharing the female pain of gendered melancholia), philanthropic mobilization for the cause of global poverty, disaster reliefs, environmental awareness, and female empowerment. Therefore, the K-pop fandom has naturally developed over the years into a socially and politically conscious collective movement, first by making these female fans vulnerable to the danger of sexism rampant in the world and then by organizing communities of female universalism that guides these fans to gender emancipation and female empowerment [11]. In a nutshell, creativity via gender, race, and postcoloniality and the consequent radical empathy are the two pillars of the sustainability of the K-pop creative industry, which make it not only economically viable, but escalate it to a level of political movements. The recent decision by some of the Arab countries to loosen regulations against K-pop and allow K-pop concerts to be held in their countries for the first time is a triumphant outcome for Arab female fans who participated in such movements [40].

Based on our discussion so far about the gendered melancholia and fan activism fecund among Hallyu and K-pop fans throughout the world, we propose the following testable hypotheses:

**Hypothesis 1 (H1):** *Gender, race, and postcoloniality among K-pop fans are positively correlated with melancholia (i.e., three types of melancholia).*

**Hypothesis 2 (H2):** *Melancholia among K-pop fans is positively correlated with the degree of their radical empathy.*

**Hypothesis 3 (H3):** *Melancholia among K-pop fans is positively correlated with the degree of Hallyu prosumption (i.e., a Tofflerian term for proactive cultural consumption).*

**Hypothesis 4 (H4):** *Hallyu prosumption is positively correlated with the degree of satisfaction with K-pop.*

**Hypothesis 5 (H5):** *Satisfaction with K-pop is positively correlated with the degree of radical empathy among K-pop fans.*

**Hypothesis 6 (H6):** *The correlation between melancholia and radical empathy among K-pop fans is mediated by Hallyu prosumption and satisfaction with Hallyu.*

**Hypothesis 7 (H7):** *The correlation between gender, race, and postcoloniality among K-pop fans and the degree of their radical empathy is mediated by melancholia, Hallyu prosumption, and satisfaction.*

The conceptual model of our current study is summarized by Figure 1 below.

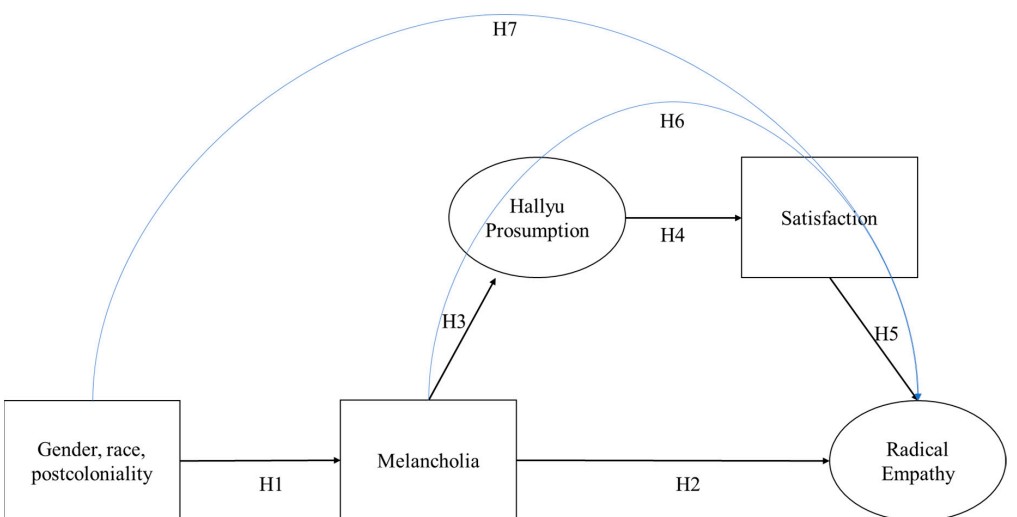

**Figure 1.** Sustainability model of melancholia and radical empathy in the K-pop creative industry.

## 3. Methodology and Analysis

### 3.1. Sample

In order to carry out the task of statistical testing of the above hypotheses, we first conducted a pilot survey of 20 participants, followed by a full-scale online survey between May and August 2021. Using a specially created website, we uploaded survey questions in English, French, Spanish, Italian, Chinese and Japanese, while we advertised the survey event on relevant fandom sites, popular social media sites, and student networks in the U.K., France, Italy, North America, Latin America, Taiwan, and Japan. We selected these countries to represent most of the major continents where K-pop is widely discussed in

ordinary mainstream discourses, either on social media or in the traditional mass media. We used a snowballing sampling method, although some of the surveys were posted on student networks in some countries making our sampling not totally random. Nonetheless, the overall sampling design emulated other large samples (e.g., ARMY fans), where students are the dominant group.

The questionnaires were first written in English and then translated into five different languages by graduate students who were bilingual in English and their mother tongue. We checked the validity of these translations through the method of back translations. In order to reduce discrepancies between the original rendition and the back translations, we repeated the process until all inconsistencies became negligible [41,42]. Among the 251 responses we received, we ruled out incomplete, inconsistent, and contradictory answers. The final sample size is consisted of 218 respondents from 15 different countries (see Table 1).

**Table 1.** Sample characteristics.

| Variable | | N | % |
|---|---|---|---|
| Nation | Afghanistan | 1 | 0.46 |
| | Argentina | 1 | 0.46 |
| | Belgium | 3 | 1.38 |
| | Canada | 14 | 6.42 |
| | Chile | 97 | 44.50 |
| | Colombia | 2 | 0.92 |
| | Denmark | 1 | 0.46 |
| | France | 27 | 13.39 |
| | Iceland | 1 | 0.46 |
| | Italy | 3 | 1.38 |
| | Japan | 28 | 12.84 |
| | Mexico | 1 | 0.46 |
| | Romania | 1 | 0.46 |
| | Taiwan | 13 | 5.96 |
| | United Kingdom | 8 | 3.67 |
| | United States | 17 | 7.80 |
| Migration | 0~2 countries | 209 | 95.87 |
| | More than 3 countries | 9 | 4.13 |
| Gender | Male | 53 | 24.31 |
| | Not Male | 165 | 75.69 |
| Sexuality | Straight | 159 | 72.94 |
| | Not Straight | 59 | 27.06 |
| Race | White | 49 | 22.48 |
| | Not White | 169 | 77.52 |
| Education | Secondary Education | 24 | 11.01 |
| | Tertiary Education | 125 | 57.34 |
| | Postgraduate Degrees | 60 | 27.52 |
| | Doctoral Degrees | 9 | 4.13 |
| Income | Unemployed/Student | 106 | 48.62 |
| | $10,000–$30,000 | 76 | 34.86 |
| | $30,000–$50,000 | 17 | 7.80 |
| | $50,000–$70,000 | 8 | 3.67 |
| | $70,000–$90,000 | 3 | 1.38 |
| | $90,000 and above | 8 | 3.67 |
| Variable | | Mean | SD |
| Age | | 27.30 | 8.84 |
| Melancholia | | 32.12 | 8.65 |

### 3.2. Variables

For the dependent variable, "Radical Empathy," we used survey questions drawn from Toronto Empathy Questionnaire (TEQ) with four new questions drawn from Given's concept of radical empathy. To determine the reliability of questionnaire items for "Radical Empathy", we used McDonald's Omega test, which generated a high score of 0.851, confirming the reliability of the survey items. For the independent variable, "Melancholia", we adopted the Sydney Melancholia Prototype Index (SMPI), which Parker et al. (2013) developed with a 24-item measuring set in order to identify and assess a potential subtype of depression: melancholia [43]. SMPI is considered a valid construct of measuring melancholia, as it has been translated into different languages and has successfully appraised melancholia symptoms in non-anglophile countries [44]. In our study, we only used the first half of the index in order to rule out the items intended to measure depressions that are different from melancholia. Following Parker and Spoelma (2021), we used five-scale items from "strongly disagree" to "strongly agree", and added up all itemized scores in order to represent the variable "Melancholia" [45]. The McDonald's Omega test generated a high score of 0.844 (standardized score of 0.841), confirming the reliability of the SMPI items.

For the mediating variables of "Hallyu Prosumption" and "Satisfaction with K-pop", we constructed survey questions based on fandom studies [46]. The satisfaction questionnaire was a single item of "how would you rate your overall experience as a Hallyu fan?" The McDonald's Omega test generated a high score of 0.901, confirming the reliability of the questionnaire items for "Hallyu Prosumption".

Finally, we added control variables to our model in order to exclude influences other than melancholia on radical empathy, such as age, postcoloniality, gender, sexuality, race, education, and employment status. As we discussed above, race, postcoloniality (i.e., one's possible exposure to postcolonialism, measured by the number of migration experiences) and gender, and gender/sexuality are thought to construct different types of melancholia (e.g., gendered, racial and postcolonial). Age was considered as a continuous variable with a range from 18 to 60, while education was divided into the three categories of secondary, university and postgraduate. Occupation was also classified into unemployed and/or student, partially employed with an income bracket of $10,000–$30,000 and fully employed with an income bracket of over $30,000. Postcoloniality was constructed to include respondents from ex-colonies (e.g., Afghanistan, Argentina, Chile, Columbia, Romania, Taiwan) and non-white respondents from ex-imperialist countries in the sample (see Table 1).

### 3.3. Method

This study utilized descriptive statistical analysis, factor analysis, and a structural equation modeling to test each hypothesis. For the explorative factor analysis (EFA), we used 0.4 as the threshold for each factor loading. For the confirmatory factor analysis (CFA), we used normed fit index (NFI), Tucker-Lewis Index (TLI), comparative fit index (CFI), composite reliability (CR), average variance extracted (AVE), and factor loadings (FL) to confirm the model fitness. We took 0.8 or above as acceptable for NFI, TLI, and CFI, and 0.7 or above for CR, while it was 0.5 or above for AVE and 0.5 or above for FL. Lastly, we tested the direct and mediating effects of selected variables using a structural equation modeling. Bootstrapping was used to measure the statistical significance of indirect effects. We used STATA 17.0 for descriptive statistics and EFA, whereas AMOS 27 was used for CFA and structural modeling.

## 4. Results and Findings

### 4.1. Descriptive Statistical Analysis

Tables 1 and 2 conjointly explain the characteristics of our sample and provide an overall result of the descriptive statistical analysis. According to both, it is clear that gender is the only control variable that is statistically significant. In our sample at least, as predicted by previous studies of K-pop [47], the female gender is a stronger predictor of melancholia than the male

gender. Furthermore, heterosexuals suffer less from melancholia than the respondents of other sexualities, as was predicted by previous studies on melancholia [11,28]. Two other control variables of "Education" and "Income" present new insights in the study of melancholia, as respondents with a higher educational achievement and income levels suffer less from melancholia than those with lower levels.

**Table 2.** Descriptive statistics for "Melancholia".

|  | Variable | N | Mean | SD | *p*-Value |
|---|---|---|---|---|---|
| Migration | 0~2 | 209 | 32.31 | 8.28 | 0.124 |
|  | More than 3 | 9 | 27.78 | 9.56 |  |
| Gender | Male | 53 | 29.66 | 7.93 | 0.017 |
|  | Not Male | 165 | 32.92 | 8.74 |  |
| Sexuality | Straight | 159 | 31.30 | 8.70 | 0.021 |
|  | Not Straight | 59 | 34.34 | 8.18 |  |
| Race | White | 49 | 30.76 | 9.24 | 0.209 |
|  | Not White | 169 | 32.52 | 8.46 |  |
| Education | Secondary Education | 24 | 33.21 | 8.70 | 0.020 |
|  | Tertiary Education | 125 | 32.67 | 7.83 |  |
|  | Postgraduate Degrees | 60 | 31.83 | 9.93 |  |
|  | Doctoral Degrees | 9 | 22.56 | 6.60 |  |
| Income | Unemployed/Student | 106 | 33.08 | 8.32 | 0.118 |
|  | $10,000–$30,000 | 76 | 32.24 | 8.74 |  |
|  | $30,000–$50,000 | 17 | 30.59 | 9.86 |  |
|  | $50,000–$70,000 | 8 | 27.38 | 7.71 |  |
|  | $70,000–$90,000 | 3 | 22.00 | 3.61 |  |
|  | $90,000 and above | 8 | 30.26 | 9.16 |  |

Note: *p*-values were calculated for "Education" and "Income" as discrete variables. All others are *t*-test results.

This test partially corroborates H1 that gender is correlated positively with melancholia (i.e., gendered melancholia), although two other variables of "Race" and "Postcoloniality" (e.g., migration from former colonies to colonizer countries) turned out to be insignificant for our sample.

### 4.2. Factor Analysis

Variables that need both exploratory and confirmatory factor analyses are: "Hallyu Prosumption" and "Radical Empathy". The exploratory factor analysis (EFA) for "Hallyu Prosumption" shows that all the survey questionnaire items, other than L01, turned out to be valid, as their factor loading scores were higher than 0.4 (see Table 3). Since the total variance explained (TVE) for all the factor loadings was above 1.00, we ruled that the EFA for "Hallyu Prosumption" is valid.

The confirmatory factor analysis for "Hallyu Prosumption" based on the explored factors produced the following result, as show in Table 4. Item A08 was excluded from the table for its lower score than 0.5. The analysis shows that CR was higher than the threshold point of 0.7, while all of NFI, TLI, and CFI turned out to be within the range of acceptable numbers. Although AVE was lower than 0.5, a generally accepted threshold point, we ruled the entire model acceptable given other pieces of swaying evidence.

**Table 3.** Hallyu Prosumption: EFA.

| | Items | Factor Loading |
|---|---|---|
| A01 | How many hours do you spend a day listening to K-pop and/or watching K-drama? | 0.741 |
| A02 | How many K-pop albums and/or K-drama DVDs have you collected so far? | 0.696 |
| A03 | How much do you spend on Hallyu a month roughly? | 0.788 |
| A04 | I have to buy a Hallyu concert ticket even if I cannot afford it. | 0.752 |
| A05 | I cannot stand without posting my comments about Hallyu every day on social media. | 0.741 |
| A06 | I have to attend local fan meetings no matter what. | 0.614 |
| A07 | I cannot survive a day without talking online about Hallyu with my fan club friends. | 0.602 |
| A08 | I always want to invite my friends who are not Hallyu fans to Hallyu fan communities. | 0.678 |
| A09 | How many fan clubs are you a member of? | 0.652 |
| L01 | How long have you been a Hallyu fan? | 0.382 |
| L02 | How would you rate yourself as a Hallyu fan? | 0.696 |
| L03 | How would you rate your loyalty to your idols? | 0.788 |
| L04 | I cannot survive a single day without Hallyu. | 0.752 |
| | Total Variance Explained (TVE) | 5.897 |

**Table 4.** Hallyu Prosumption: CFA.

| | Items | Factor Loading | McDonald's Omega | CR | AVE |
|---|---|---|---|---|---|
| | A01 | 0.743 | 0.901 | 0.899 | 0.451 |
| | A02 | 0.614 | | | |
| | A03 | 0.602 | | | |
| | A04 | 0.681 | | | |
| | A05 | 0.646 | | | |
| Hallyu Prosumption | A06 | 0.591 | | | |
| | A07 | 0.665 | | | |
| | A09 | 0.565 | | | |
| | L02 | 0.681 | | | |
| | L03 | 0.795 | | | |
| | L04 | 0.762 | | | |
| CMIN = 181.123/DF = 44/RMSEA = 0.120/NFI = 0.841/TLI = 0.841/CFI = 0.873 | | | | | |

The same procedures were repeated for radical empathy. Excluding all the items that did not pass the barrier of 0.4, we explored 11 items that satisfactorily measured the variable (Table 5). Since TVE for all the factor loadings was above 1.00, we ruled that the EFA for radical empathy is valid. The CFA for the variable is shown in Table 6, which has already excluded items that had lower than 0.5 for factor loadings. CR has yielded 0.851 above the barrier of 0.7, while NFI and CFI cleared the hurdle of 0.8. Although TLI and AVE were lower than 0.8 and 0.5, respectively, we ruled this model to be acceptable based on other promising indicators.

**Table 5.** Radical Empathy: Explorative Factor Analysis.

| | Items | Factor Loading |
|---|---|---|
| E01 | When someone else is feeling excited, I tend to get excited too. | 0.366 |
| E02 * | Other people's misfortunes do not disturb me a great deal | −0.178 |
| E03 | It upsets me to see someone being treated disrespectfully. | 0.599 |
| E04 * | I remain unaffected when someone close to me is happy. | −0.377 |
| E05 | I enjoy making other people feel better. | 0.549 |
| E06 | I have tender, concerned feelings for people less fortunate than me. | 0.599 |
| E07 * | When a friend starts to talk about his/her problems, I try to steer the conversation towards something else. | −0.293 |
| E08 | I can tell when others are sad, even when they do not say anything. | 0.432 |
| E09 | I find that I am "in tune" with other people's moods. | 0.479 |
| E10 * | I do not feel sympathy for people who cause their own serious illnesses. | −0.290 |
| E11 | I become irritated when someone cries. | −0.311 |
| E12 * | I am not really interested in how other people feel. | −0.390 |
| E13 | I get a strong urge to help when I see someone who is upset. | 0.604 |
| E14 * | When I see someone being treated unfairly, I do not feel very much pity for them. | −0.260 |
| E15 * | I find it silly for people to cry out of happiness. | −0.319 |
| E16 | When I see someone being taken advantage of, I feel kind of protective towards him/her. | 0.640 |
| E17 | If I believe I have to help someone, I am willing to publicly expose my identity even though that would make myself vulnerable and put myself in danger. | 0.531 |
| E18 | I have no reservation in joining a social movement that would improve my and others' social and economic equality. | 0.560 |
| E19 | I have no reservation in donating money for the cause of a social movement that I believe would make this world better. | 0.546 |
| E20 | I am willing to help others who are economically worse off than me. | 0.597 |
| | TVE | 5.077 |

* Represents reverse value coding items.

**Table 6.** Radical Empathy: CFA.

| | Items | Factor Loading | McDonald's Omega | CR | AVE |
|---|---|---|---|---|---|
| | E03 | 0.611 | 0.851 | 0.851 | 0.391 |
| | E05 | 0.561 | | | |
| | E06 | 0.630 | | | |
| | E13 | 0.527 | | | |
| | E16 | 0.679 | | | |
| Radical Empathy | E17 | 0.608 | | | |
| | E18 | 0.625 | | | |
| | E19 | 0.662 | | | |
| | E20 | 0.702 | | | |
| | E03 | 0.611 | | | |
| | E05 | 0.561 | | | |
| CMIN = 128.918/DF = 27/RMSEA = 0.132/NFI = 0.812/TLI = 0.791/CFI = 0.843 | | | | | |

### 4.3. Structural Equation Modeling

Finally, our structural equational modeling and bootstrap analysis corroborate both direct and indirect effects of all the contested variables in the model (Table 7). While

gendered melancholia (H1) turned out to have direct effects on radical empathy (H2), the correlation between gender and radical empathy was mediated by melancholia, Hallyu prosumption, and satisfaction (H7). The mediating effect of Hallyu prosumption and satisfaction was also supported between melancholia and radical (H6). Finally, melancholia, Hallyu prosumption, satisfaction, and radical empathy all had direct effects on each other (H3, H4, H5).

**Table 7.** Bootstrap analysis (structural equation modeling).

| | | | Bias Corrected Confidence Interval | |
| --- | --- | --- | --- | --- |
| **Path** | **Effects (Standardized)** | **$p$-Value** | **Lower** | **Upper** |
| H1: Gender → ME | 2.960 (0.148) | 0.027 | 0.915 | 5.151 |
| H1: Race → ME | 0.674 (0.033) | 0.625 | −1.803 | 3.216 |
| H1: Migration → ME | −3.523 (−0.081) | 0.223 | −9.376 | 1.596 |
| H2: ME → RE | 0.014 (0.160) | 0.027 | 0.003 | 0.026 |
| H3: ME → HP | 0.038 (0.273) | 0.000 | 0.022 | 0.055 |
| H4: HP → SA | 0.691 (0.604) | 0.003 | 0.564 | 0.815 |
| H5: SA → RE | 0.121 (0.218) | 0.000 | 0.059 | 0.179 |
| H6: ME → PH → SA → RE | 0.003 (0.036) | 0.001 | 0.001 | 0.006 |
| H7: Gender → ME → PH → SA → RE | 0.052(0.029) | 0.024 | 0.014 | 0.118 |

The findings so far can be synthesized into the final model with the test results as follows (Figure 2):

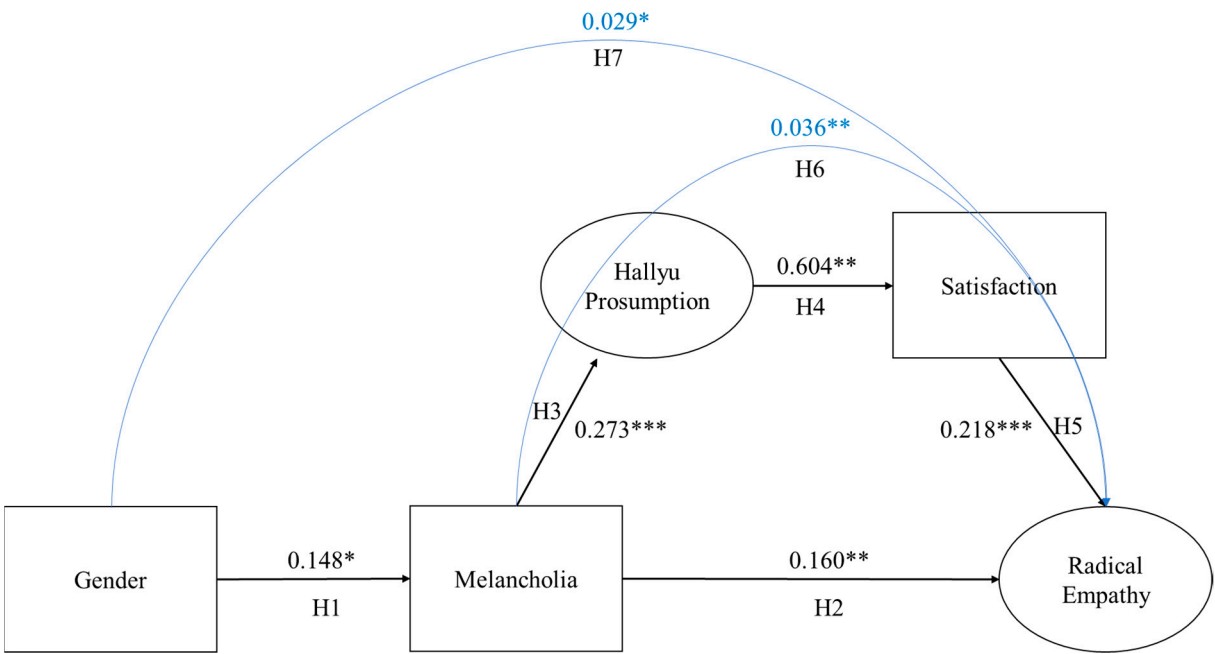

**Figure 2.** "Melancholia" and "Radical Empathy" with results. Note: * ($p \leq 0.05$), ** ($p \leq 0.01$), and *** ($p \leq 0.001$).

## 5. Discussion

As Figure 2 shows, all seven hypotheses that have been corroborated against the survey data we collected represent different areas of significance in the multidisciplinary study of the creative industries. Table 8 below summarizes our findings and their significance.

**Table 8.** Findings and significance.

| Hypotheses | Dimensions and Disciplines | Significance of Findings |
|---|---|---|
| H1 | Melancholia (feminist philosophy, psychiatry) | The source of melancholia confirmed; gender looms large for our sample. Further studies needed to confirm the significance of race and postcoloniality. |
| H2, H5 | Radical empathy (sociology, social work, cultural studies, fandom studies) | The source of radical empathy confirmed; gender and satisfaction loom large for our sample. Further studies needed to find other sources. |
| H3, H4 | Prosumption (cultural studies, fandom studies, marketing studies) | The source of prosumption confirmed; gendered melancholia looms large for our sample. Prosumption also became a source of consumption satisfaction in our sample. Further studies needed to confirm other sources. |
| H6 | Melancholia and radical empathy (Feminist philosophy, psychiatry) | The mediating factors between melancholia and radical empathy confirmed; prosumption and satisfaction loom large in our sample. Further studies needed to confirm other sources. |
| H7 | Gender and radical empathy (sociology, cultural studies, feminist philosophy, psychiatry) | The mediating factors between gender and radical empathy confirmed; melancholia, prosumption, and satisfaction loom large in our sample. Further studies needed to confirm other sources. |

As can be seen in Table 8, we contribute to important findings across a wide range of areas and subjects in the social sciences. These vary from not only cultural studies and sociology, but also marketing, philosophy, and stretching even to psychiatry. These shed light and insights on areas, such as diversity, regarding gender and race, as well as consumption patterns. Furthermore, our findings show the importance of the need for more critical thinking and moving perspectives, methods, and analyses beyond the narrow confines of "silos", to more multidisciplinary approaches. As our findings indicate, our research strategy seems valuable when it comes to complex areas such as studying the cultural and/or creative industries.

As problems and issues in the world grow ever more complex, the need to shift away from narrow to broader perspectives in order to understand and make an impact becomes more pressing. This has been recognized in a range of areas, such as international business [48]. There are thorny issues of course, such as bridging disciplines and processes [49], how to analyze interdisciplinarity in terms of typologies and indicators [50], where to publish such work, and how it is reviewed and evaluated, etc. Despite difficulties and limitations, which we will explain below, our current approach has gained new insights about the sustainability of creative industries from a diverse range of theoretical perspectives. By being able to pin down the source of creativity and linking them with the self-realization of individual creativity in the new middlebrow art genres, the multidisciplinary, or what some may call "transdisciplinary", strategy could save the once theoretically and morally bankrupt concept of cultural and creative industries [51–54].

*5.1. Limitations*

This study has limitations. First, the sample size should be larger than the current one, with more countries to cover, so that it can be highly randomized in the sampling process. In addition, the operationalization of melancholia and radical empathy needs to be further developed with more indexing tools. For example, the current measure fails to distinguish between gendered from racial or postcolonial melancholia. Furthermore, the measures of melancholia and radical empathy must be culturally sensitive, as some questions are too English-centered, causing semantic difficulties when translated into different languages with very different cultural backgrounds. Finally, this study provides no reliable measure of sustainability, as the concept was not operationalized into a measurable variable. Instead, it was simply assumed to be highly correlated with prosumption and radical empathy. One less significant limitation than others may be the lack of our discussion on the regional and national variations of postcolonial creativity that is relevant to gendered, racial, and

postcolonial melancholia. However, the purpose of our paper is to establish how melancholia can serve as a pivot for creativity and sustainability of postcolonial middlebrow art for the first time in our discipline. Discussions of varieties of postcolonial creativity and sustainability can be presented in other research outputs.

### 5.2. Implications and Directions for Future Research

The findings of this study provide a wide range of theoretical and practical implications. Theoretically, the empirical evidence presented in this paper supports the creative industry theory, which tries to rescue the old notion of the cultural industries from its conceptual bankruptcy [55]. The age-old criticism of the cultural industries from Marxists, Frankfurt School scholars, and recent postmodernists, that the entire business value chain that produces and distributes cultural products in mass quantities has succumbed to the desire of the financial capital, can now be seriously challenged [1–5]. First and foremost, consumers of cultural products are not mere passive followers of the top-down messages inculcated in the cultural products in an Orwellian fashion, as they are proactive selectors and promoters of particular social and political messages embedded in each cultural product that they carefully select to consume (i.e., prosumption and prosumers). Furthermore, the source of creativity rampant in the creation of the middlebrow cultural genres, that are not only innovative in creating aesthetic content, but sustaining the entire industry, as well through high profitability, now looms large [6,7]. Moreover, the psychiatric studies that have successfully linked melancholia with creativity in innovative and avant-garde art has also supported this new view of the creative industries and their sustainability [33–37]. Therefore, the key to the sustainability of creative industries is not only in the existence of such creativity per se, but the expansion of creativity into radical empathy that would enhance consumers' participation in the cultural, social, and political movements that would help sustain the entire industry by strengthening their cultural empathy towards minority groups of the society [13,56].

Practically, this study provides several implications. First, firms in the creative industries, especially in postcolonial societies, can utilize the findings and implications of this study. They can learn how to package their cultural products that can lead to cultural prosumption and radical empathy. In order to do that, firms should mobilize creative resources (e.g., gender, race, postcoloniality) in designing, producing, and distributing their cultural goods and services. Second, firms in creative industries must utilize transnational resources of creativity that defies nationalism and geographical boundaries. To achieve this, firms should flout nationalism as their core message, a popular ideology within the cultural industries in the previous century, and instead adopt a new strategy of cultural transnationalism that can unify creative resources globally under the banner of gender, race, and postcoloniality. Finally, firms should nurture communities of prosumption and radical empathy, not the communication devices and applications that simply promote mass consumption. Firms in creative industries are not gate keepers in Hirsch's sense of the cultural industries with high entry barriers [8], but a community builder, with transnational orientation toward incorporating larger groups with diverse gender, race, and postcolonial identities. In this sense, entry barriers into new creative industries are substantially lower than those in the cultural industries.

Future studies, therefore, need to develop novel theories and methods that can propose an alternative understanding of the creative industries and their sustainability to those readily available for the outdated cultural industries with banners of nationalism, divisions, and discriminations. New studies also can envisage a historical and comparative research design to discuss the geographical and historical specificities of the creative industries with their new characteristics of creativity, prosumption, and radical empathy.

### 6. Conclusions

This is the first study that quantitatively analyzes the source of creativity and sustainability in the creative industry that is dominated by non-white, non-European, or non-North

American artists. It finds that, at least within the K-pop community, gender (or gendered melancholia among females) is paramount to musical creativity in the middlebrow artistic genre, that is both highly aesthetic and profitable. Key to the success of this creative industry as an ecological whole is the hidden mechanism of gendered melancholia that promotes radical empathy through art prosumption and its satisfaction. Radical empathy, in turn, empowers females in the K-pop community all over the world to orchestrate and organize social and political movements, either physically or digitally, in order to make the world more sustainable and friendly than ever for women, as well as men, while promoting human and environmental justice.

The question of the artistic creativity and its innovative orientation toward imagining a middlebrow art genre, pivotal to the sustainability of the creative industries, is benefited by sociology (e.g., the discovery of gender, race, and postcoloniality in artistic innovation), feminist philosophy (e.g., the discovery of gendered melancholia for female creativity and empowerment), psychiatry (e.g., the discovery of the nexus between depression and creativity), and business studies (e.g., the discovery of the creative industries vis-à-vis the outdated notion of the cultural industries).

This vast body of knowledge regarding creativity, gender, race, postcoloniality, melancholia, and creative industries reveals new possibilities of the creative industries that can be saved from both Marxist and postmodernist critique, that tend to consider it as a new form of the old machine of mass brainwashing, or as an old form of male domination in an ever diversifying and hybridizing field of global cultural business. This study, therefore, helps continue the discussion of creative industries and their sustainability in a balanced and objective fashion.

**Author Contributions:** Conceptualization, I.O.; methodology, K.-J.K.; writing—original draft preparation, I.O.; writing—review and editing, C.R.; visualization, K.-J.K. and I.O. All authors have read and agreed to the published version of the manuscript.

**Funding:** This research received no external funding.

**Institutional Review Board Statement:** Not applicable.

**Informed Consent Statement:** Consent Form available upon request.

**Data Availability Statement:** Not applicable.

**Acknowledgments:** An earlier version of this paper was presented at the 2021 Meeting of East Asian Sociological Association. We thank Carolina Pérez Núñez, Sheraze Maazi, Stefania Pozzi, Hsingyu Cheng, and Terumi Nakano for helping us translate surveys into their mother tongues and posting them in various online sites. We also thank Lynn Pyun for her comments on our paper.

**Conflicts of Interest:** The authors declare no conflict of interest.

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
