# Peer review of "Female Empowerment and Radical Empathy for the Sustainability of Creative Industries: The Case of K-Pop"

_sustainability, doi:10.3390/su15043098_

Round 1
Reviewer 1 Report
The article is interesting and uses in an original way the example of Kpop to answer broader questions concerning the transformations of "artist" capitalism, the feminization of the public and its explanatory dynamics as well as the elaboration of moral links.
a) It seems to me that the mobilized literature could be enriched and updated (because the references of the theoretical frameworks are a bit old) on each of these points in order to propose a less monolithic reading of the phenomena:
-on the transformations of capitalism turned artist and the place of creativity: the Marxist analyses, if they remain stimulating, are completed by new other contributions, which should be put into perspective. See Murphy& de la Fuente, Böhme, Featherstone
-on the uses of emotions in this capitalism: see E. Illouz
-On the links between feelings and fandom: see J. Radway, H. Wood, R. Collins, etc.
b) The link between melancholy and creativity seems to me to need to be worked on more precisely because it seems to me that the authors use it as a causal relationship (melancholy would "create" creativity whereas it is rather a co-occurrence in certain cases. In other words, some creators were affected by melancholia, but not all by far, and conversely, not all patients with melancholia are creative. It would also be necessary to better distinguish professional artistic creativity from its more banal, everyday and widespread forms. Creation and creativity are not synonymous.
c) We must also beware of misleading generalizations: the social conditions of melancholia - which vary from country to country - resonate with the conditions of women - which also vary: it seems strange to me to appeal to a universal feeling without discussing its local variations (or even its local distortions)
The presentation of the statistical models is clear, even if the number of hypotheses tested could be reduced because some of them are very related.
It would benefit from further discussion of the suggested authors and especially from discussion of the fact that the notion of a melancholic universal, proposed by the authors, contradicts readings of the construction of gender (and its consequences) in terms of a socially and historically situated construction...., readings which are in the majority in gender studies.
Author Response
Author Responses
Thanks for reviewing our paper in a swift manner with full of insights and constructive comments. We have revised our paper according to your feedback. The following is how we dealt with each problem.
Reviewer 1:
The article is interesting and uses in an original way the example of Kpop to answer broader questions concerning the transformations of "artist" capitalism, the feminization of the public and its explanatory dynamics as well as the elaboration of moral links.
- Thanks for your complimentary comment.
- a) It seems to me that the mobilized literature could be enriched and updated (because the references of the theoretical frameworks are a bit old) on each of these points in order to propose a less monolithic reading of the phenomena:
-on the transformations of capitalism turned artist and the place of creativity: the Marxist analyses, if they remain stimulating, are completed by new other contributions, which should be put into perspective. See Murphy& de la Fuente, Böhme, Featherstone
-on the uses of emotions in this capitalism: see E. Illouz
-On the links between feelings and fandom: see J. Radway, H. Wood, R. Collins, etc.
- Thanks for these valuable sources. We have incorporated most of these sources in our literature review. However, our main argument is not affected by these new sources.
- b) The link between melancholy and creativity seems to me to need to be worked on more precisely because it seems to me that the authors use it as a causal relationship (melancholy would "create" creativity whereas it is rather a co-occurrence in certain cases. In other words, some creators were affected by melancholia, but not all by far, and conversely, not all patients with melancholia are creative. It would also be necessary to better distinguish professional artistic creativity from its more banal, everyday and widespread forms. Creation and creativity are not synonymous.
- This problems has never really been intended by us, and we think it was our English problem as non-native speakers. We have thoroughly corrected any misleading words that may sound like a causal relationship between the two variables.
- c) We must also beware of misleading generalizations: the social conditions of melancholia - which vary from country to country - resonate with the conditions of women - which also vary: it seems strange to me to appeal to a universal feeling without discussing its local variations (or even its local distortions)
- The point you raise here is absolutely correct. But our intention is not to discuss how varieties of creativity exists in the world. Our intention is to show there is a universal source of creativity and addiction within the community of K-pop (or postcolonial pop) culture. After establishing the importance of race, gender, and postcoloniality in postcolonial pop culture industries, we can then discuss varieties and local transformations. This is a separate task from the current one.
The presentation of the statistical models is clear, even if the number of hypotheses tested could be reduced because some of them are very related.
- Thanks for this beautiful insight. However, since this is also a paper in business studies, we did not heed, as in business it is very common to present several hypotheses separately although they’re related. So, please understand this problem in an interdisciplinary context.
It would benefit from further discussion of the suggested authors and especially from discussion of the fact that the notion of a melancholic universal, proposed by the authors, contradicts readings of the construction of gender (and its consequences) in terms of a socially and historically situated construction...., readings which are in the majority in gender studies.
- Thanks for this constructive comment, which we much do in our next project. But for now we want to show that previous gender studies failed to explain the gendered phenomenon and its global success of K-pop. Therefore, we did not want to incorporate gender studies in our paper. If you know any convincing gender studies that explain K-pop, please share the sources so that we can incorporate them in our current study.
Reviewer 2 Report
The paper is well-written but it is somewhat complicated and thus not clear. Especially, it does not prove any relationship between gender melancholia (this word, although seems to be central in this paper,, it does not mentioned in the title) and sustainability or sustainable development.
The authors investigate a complicated relationship between melancholia (due to gender sexism), radic empathy (who feels empathy for whom), then innovation (innovation considered a result of melancholia), economic profits (in this paper economic efficiency and profit-making is considered to have a potential for sustainability) and finaly sustainability itself (not clearly explained how this set-up is sustainable)
Moreover, although I am not an expert in creative industry the authors put in the same framework well-known music genres such as Jazz, Rock and k-pop in order to state (as far as I understand) that k-pop is 'sustainable' somewhat like by definition, interpreting sustainability as long-run existence)
Also, the wording in the case of other movements (e.g. demoralized commercial activities) is disparaging. The insertion of money in economic setups does not make the models demoralized.
Concluding my review, and having read another paper from the same authors (i.e. gendered melamcholia as cultural branding) I think that in order for this paper to be robust, the authors have to first state what they excactly mean by using the word sustainable in their setup and then explain clearly how all this psychological setup creates empathy, how empathy creates innovation and creativity, and finally how all these lead to sustainable development. Ideally, in order for this to get explained clearly, sustainabilty (in the form of long-term equilibria) should be an endogenous variable in the model. Otherwise, gendered melancholia itself do not explain any sustainable perspective.
Author Response
Author Responses
Thanks for reviewing our paper in a swift manner with full of insights and constructive comments. We have revised our paper according to your feedback. The following is how we dealt with each problem.
Reviewer 2:
The paper is well-written but it is somewhat complicated and thus not clear. Especially, it does not prove any relationship between gender melancholia (this word, although seems to be central in this paper, it does not mentioned in the title) and sustainability or sustainable development.
- Thanks for your comment. We tried to change our wording and sentences to clearly show the relationship between gender melancholia and sustainability. In other words, the key concept is that there is a tangible and sustainable reason why K-pop is a global success (i.e., longevity). The key factor is gender, race, and postcolonial melancholia both among the artists and the fans. Furthermore, they have formed a community of radical empathy that would unite them together as a strong fandom movement (e.g., the ARMY of BTS) and prolong the longevity, thus sustainability, of the postcolonial pop culture, such as K-pop. Hope this is clear now.
The authors investigate a complicated relationship between melancholia (due to gender sexism), radical empathy (who feels empathy for whom), then innovation (innovation considered a result of melancholia), economic profits (in this paper economic efficiency and profit-making is considered to have a potential for sustainability) and finaly sustainability itself (not clearly explained how this set-up is sustainable)
- We have deleted any discussion/mentions about environmental sustainability in our paper, although the K-pop fandom movement include pro-environmental movement in it. In this paper, sustainability is purely economic.
Moreover, although I am not an expert in creative industry the authors put in the same framework well-known music genres such as Jazz, Rock and k-pop in order to state (as far as I understand) that k-pop is 'sustainable' somewhat like by definition, interpreting sustainability as long-run existence)
- K-pop’s been around more than 25 years, and it is becoming bigger than Jazz and classic Rock in terms of fandom and global annual revenues. Therefore, it is not totally unreasonable to parallel K-pop with Jazz and Rock. Hope this makes sense to you.
Also, the wording in the case of other movements (e.g. demoralized commercial activities) is disparaging. The insertion of money in economic setups does not make the models demoralized.
- Thanks for the comment. We have changed the wording and deleted morality or immorality in our paper.
Concluding my review, and having read another paper from the same authors (i.e. gendered melancholia as cultural branding) I think that in order for this paper to be robust, the authors have to first state what they exactly mean by using the word sustainable in their setup and then explain clearly how all this psychological setup creates empathy, how empathy creates innovation and creativity, and finally how all these lead to sustainable development. Ideally, in order for this to get explained clearly, sustainability (in the form of long-term equilibria) should be an endogenous variable in the model. Otherwise, gendered melancholia itself do not explain any sustainable perspective.
- Thanks for the comment. We have clearly defined sustainability as purely an economic concept (e.g., profit making on a long term basis). Although we did not incorporate the sustainability variable in our model, our intention is to analyze the black box of creativity and radical empathy within the postcolonial pop culture industry, both of which are pivotal in making K-pop economically sustainable. The economic success of K-pop is given and assumed in our paper, and this is not unusual in the discipline. Hope this reduces your worry.
Round 2
Reviewer 1 Report
The paper has improved. There is still discussion to be done regarding concepts and materials but it will maybe the motivation for another paper
Reviewer 2 Report
Empathy as a source of innovation is a well-known topic in human-centric design. In this paper, empathy takes the form of female pain of gendered melancholia. Therefore, melancholia could be considered and seen in a new framework; a combination of melancholia and empathy that serves the needs of the youth, especcially female.
I would like to see this effort about k-pop as an economically efficient movement for achieving Goals 5 and 10 of UNESCO SDGs through art.